# Inhibitor-induced HER2-HER3 heterodimerisation promotes proliferation through a novel dimer interface

Jeroen Claus[1†], Gargi Patel[2,3†], Flavia Autore[4], Audrey Colomba[1], Gregory Weitsman[2], Tanya N Soliman[1], Selene Roberts[5], Laura C Zanetti-Domingues[5], Michael Hirsch[5], Francesca Collu[4], Roger George[6], Elena Ortiz-Zapater[7], Paul R Barber[4,8], Boris Vojnovic[4,9], Yosef Yarden[10], Marisa L Martin-Fernandez[5], Angus Cameron[1,11]*, Franca Fraternali[4]*, Tony Ng[2,8,12]*, Peter J Parker[1,13]*

[1]Protein Phosphorylation Laboratory, The Francis Crick Institute, London, United Kingdom; [2]Richard Dimbleby Department of Cancer Research, Randall Division and Division of Cancer Studies, Kings College London, London, United Kingdom; [3]Sussex Cancer Centre, Brighton and Sussex University Hospitals, Brighton, United States; [4]Randall Division of Cell & Molecular Biophysics, Kings College London, London, United Kingdom; [5]Central Laser Facility, Research Complex at Harwell, Science and Technology Facilities Council, Rutherford Appleton Laboratory, Didcot, United Kingdom; [6]The Structural Biology Science Technology Platform, The Francis Crick Institute, London, United Kingdom; [7]Department of Asthma, Allergy and Respiratory Science, King's College London, Guy's Hospital, London, United Kingdom; [8]UCL Cancer Institute, University College London, London, United Kingdom; [9]Department of Oncology, Cancer Research UK and Medical Research Council Oxford Institute for Radiation Oncology, Oxford, United Kingdom; [10]Department of Biological Regulation, Weizmann Institute of Science, Rehovot, Israel; [11]Barts Cancer Institute, Queen Mary University of London, London, United Kingdom; [12]Breast Cancer Now Research Unit, Department of Research Oncology, Guy's Hospital King's College London School of Medicine, London, United Kingdom; [13]School of Cancer and Pharmaceutical Sciences, King's College London, Guy's Campus, London, United Kingdom

*For correspondence:
a.cameron@qmul.ac.uk (AC);
franca.fraternali@kcl.ac.uk (FF);
tony.ng@kcl.ac.uk (TN);
peter.parker@crick.ac.uk (PJP)

†These authors contributed equally to this work

Competing interests: The authors declare that no competing interests exist.

**Abstract** While targeted therapy against HER2 is an effective first-line treatment in HER2[+] breast cancer, acquired resistance remains a clinical challenge. The pseudokinase HER3, heterodimerisation partner of HER2, is widely implicated in the resistance to HER2-mediated therapy. Here, we show that lapatinib, an ATP-competitive inhibitor of HER2, is able to induce proliferation cooperatively with the HER3 ligand neuregulin. This counterintuitive synergy between inhibitor and growth factor depends on their ability to promote atypical HER2-HER3 heterodimerisation. By stabilising a particular HER2 conformer, lapatinib drives HER2-HER3 kinase domain heterocomplex formation. This dimer exists in a head-to-head orientation distinct from the canonical asymmetric active dimer. The associated clustering observed for these dimers predisposes to neuregulin responses, affording a proliferative outcome. Our findings provide mechanistic insights into the liabilities involved in targeting kinases with ATP-competitive inhibitors and highlight the complex role of protein conformation in acquired resistance.

DOI: https://doi.org/10.7554/eLife.32271.001

## Introduction

The epidermal growth factor receptor (EGFR) family of receptor tyrosine kinases plays a major role in proliferative signalling in a variety of cancers (*Baselga and Swain, 2009*; *Yarden and Pines, 2012*). Apart from EGFR (also known as ErbB1), the family consists of the orphan receptor HER2 (ErbB2), the pseudokinase HER3 (ErbB3), and HER4 (ErbB4). Overexpression of HER2 is an oncogenic driver in approximately 20% of all breast cancers (*Lovekin et al., 1991*; *Owens et al., 2004*; *Slamon et al., 1987*). The high clinical relevance of these receptors has made them a target for directed therapy with both antibodies and small molecule kinase inhibitors. In the case of HER2$^+$ breast cancer, the monoclonal antibody trastuzumab (Herceptin) and its cytotoxic drug-conjugated derivative trastuzumab-emtansine (Kadcyla), the monoclonal antibody blocking HER2-HER3 dimerisation pertuzumab (Perjeta), and the small molecule kinase inhibitor lapatinib (Tykerb/Tyverb) have been successful in the clinic (*Blackwell et al., 2010*; *Cameron et al., 2017*; *Diéras et al., 2017*; *Geyer et al., 2006*; *Krop et al., 2017*; *CLEOPATRA Study Group et al., 2015*; *EMILIA Study Group et al., 2012*).

While HER2 itself has no known ligand, HER3 binds the growth factor neuregulin (NRG, also known as heregulin or HRG) to induce heterodimerisation and signalling (*Sliwkowski et al., 1994*). HER3 has been implicated in therapeutic resistance to HER2-targeted therapy through a variety of mechanisms, including receptor rephosphorylation, HER3 overexpression and increased NRG production (reviewed in [*Claus et al., 2014*]). In terms of cellular signalling in response to HER-family kinase inhibition, HER3-mediated buffering through the Akt/PKB signalling axis has been shown to be an important factor in therapeutic resistance (*Sergina et al., 2007*).

The dimerisation of EGFR family members is a fluid process mediated by interaction dynamics in practically every domain of the receptor. For EGFR, the ligand-bound, active dimer shows an upright, back-to-back extracellular domain (ECD) interaction where both receptors have bound ligand, although singly-bound dimers can also occur (*Garrett et al., 2002*; *Liu et al., 2012*; *Ogiso et al., 2002*). Although HER2 has no known ligand, it natively adopts this upright, dimerisation-ready ectodomain conformation (*Garrett et al., 2002*). On the intracellular side, formation of the active kinase domain dimer is critically affected by the conformation of the juxtamembrane domain (JMD) (*Jura et al., 2009a*; *Thiel and Carpenter, 2007*). The kinase domains associate in an asymmetric dimer, which resembles the CDK/cyclin-like asymmetric dimer interface (*Jeffrey et al., 1995*; *Zhang et al., 2006*). In this canonical dimer, one kinase (the 'activator') allows the dimerisation partner (the 'receiver') to adopt an active conformation and become catalytically active. These various conformations have also been observed in near-complete receptors using negative stain electron microscopy (*Mi et al., 2011*). Of note in these receptor dimer formations was the lack of active, asymmetrical kinase domain interactions when the receptor was bound to the ATP-competitive inhibitor lapatinib (*Mi et al., 2011*). Although these interactions have mainly been described in the context of EGFR homodimerisation, they remain a template for the interactions of the rest of the EGFR family. The conformation of the active kinase domain interaction has been validated for EGFR-HER3 and HER2-HER3 (*Jura et al., 2009b*; *Littlefield et al., 2014*; *van Lengerich et al., 2017*).

A multitude of studies, using a variety of techniques, have confirmed that EGFR-family receptors can form higher order oligomers, and that the exact nature of these oligomers is modulated by a variety of conditions, including receptor density, ligand presence, ligand type and temperature-dependent membrane behaviour (*Clayton et al., 2005*; *Clayton et al., 2007*; *Huang et al., 2016*; *Nagy et al., 2010*; *Needham et al., 2016*; *Saffarian et al., 2007*; *van Lengerich et al., 2017*; *Yang et al., 2007*; *Zhang et al., 2017*).

Against the backdrop of such a multitude of association modes, it is clear that conformational dynamics and structural rearrangements are an integral regulator of protein behaviour in the EGFR family.

We have shown previously that within a kinase, in this case PKCε, occupation of the nucleotide binding pocket with ATP (or an inhibitor) is a major determinant of protein behaviour, conferring the structural stability required for protein–protein interactions to occur and priming sites to be stably phosphorylated (*Cameron et al., 2009*). Similar effects have been observed in several additional

**eLife digest** Around 20% of breast cancers are caused because cells have too many copies of a receptor protein called HER2 on their surface. HER2 is responsible for telling the cell to divide. Cells with too many of these receptors – and breast cancer cells can have up to 1000 times too many – divide uncontrollably. This causes the cancer to grow.

Several successful anti-cancer drugs, such as Herceptin and Kadcyla, are used in the clinic to block the signals produced by HER2. Other drugs called kinase inhibitors prevent HER2 from building its faulty signals. However, a particular kinase inhibitor called lapatinib was not as successful in clinical trials as the medical community had hoped.

Kinase inhibitors can have unexpected effects. While they can block specific signals in a cell, they can sometimes also cause new types of signals. Could this be one of the reasons behind the disappointing clinical trial results for lapatinib?

By performing experiments on breast cancer cells grown in the laboratory, Claus, Patel et al. found that lapatinib can counterintuitively boost the growth of breast cancer cells. This occurs because lapatinib causes HER2 receptors to cluster together like a daisy chain along with another protein receptor of the same family, called HER3. These chains are primed to rapidly respond to a molecule called neuregulin, a growth factor that is commonly associated with breast cancer.

The results presented by Claus, Patel et al. indicate that a particular subset of breast cancer patients – those whose cancer cells do not increase production of HER3 receptors – might better respond to lapatinib than others. The insights gained into what happens to HER2 when you try to block it should also influence the design of new drugs that target either HER2 or HER3.
DOI: https://doi.org/10.7554/eLife.32271.002

kinases, including PKB/Akt, IRE1, and AMPK (*Okuzumi et al., 2009*; *Papa et al., 2003*; *Ross et al., 2017*; *Wang et al., 2012*).

A notable example of nucleotide binding pocket occupation inducing behaviour independent of catalysis has been described for the RAF family, originally in cRAF, where the inhibitor SB 203580 paradoxically induced activity (*Eyers et al., 1998*). More recently, a similar phenomenon has been shown in BRAF, where the small molecule kinase inhibitor vemurafenib blocks the oncogenic mutant V600E, but stabilises the wild type protein, promoting downstream proliferative signalling (*Hatzivassiliou et al., 2010*; *McKay et al., 2011*; *Poulikakos et al., 2010*; *Thevakumaran et al., 2015*). Within the EGFR family, we and others have shown previously that quinazoline inhibitors can cause homodimer formation of EGFR, and EGFR-MET heterodimerisation, by stabilising particular kinase domain conformers (*Arteaga et al., 1997*; *Bublil et al., 2010*; *Lichtner et al., 2001*; *Ortiz-Zapater et al., 2017*).

The structural, conformational role that nucleotide pocket occupation can fulfil is particularly interesting in the context of pseudokinases, which have lost their catalytic activity. Sequence analysis shows that many pseudokinases retain several of the conserved residues involved in ATP-binding (*Boudeau et al., 2006*; *Claus et al., 2013*). *In vitro* analysis of the pseudokinome showed that many pseudokinases have nucleotide binding capability (*Murphy et al., 2014*).

In the case of these ATP-binding pseudokinases, where nucleotide binding does not elicit phosphotransfer, the structural stability conferred by ATP binding may be integral to protein function. This has been observed for the pseudokinase STRAD, which requires ATP binding to sustain a heterotrimeric complex with LKB and MO25 (*Zeqiraj et al., 2009a*; *Zeqiraj et al., 2009b*). Similarly, in the pseudokinase FAM20A ATP-binding, albeit in a non-canonical orientation, is essential for stabilising the FAM20A/FAM20C complex (*Cui et al., 2015*; *Cui et al., 2017*). ATP binding is a structural requirement for the JAK2 JH2 V617F mutant to promote pathogenic signalling (*Hammarén et al., 2015*). In the pseudokinase MLKL, ATP-binding pocket occupation is essential for membrane translocation and its role in necroptotic signalling (*Hildebrand et al., 2014*; *Murphy et al., 2013*).

HER3 is able to bind ATP (crystallised as PDB ID 3KEX, 3LMG), as well as the Src/ABL inhibitor Bosutinib (PDB ID 4OTW) (*Levinson and Boxer, 2014*; *Davis et al., 2011*; *Jura et al., 2009b*; *Murphy et al., 2014*; *Shi et al., 2010*). Considering the importance of HER3 as a conformational

partner in the HER2-HER3 heterodimer, and the established importance of ATP-binding for complex formation in other pseudokinases, the role of nucleotide binding pocket occupation in HER3 function warrants investigation.

Here, we have integrated the study of kinase-autonomous conformational effects of nucleotide binding pocket occupation with that of HER2-HER3 heterointeraction modalities and downstream proliferative phenotypes in response to drug treatment. We show that nucleotide pocket occupation in both HER2 and the pseudokinase HER3 is of great conformational importance for kinase domain heterodimerisation and subsequent proliferative signalling. In HER2$^+$ breast cancer cells this leads to an unexpected synergy between the HER3 ligand NRG and the HER2 inhibitor lapatinib, by which their concomitant binding promotes proliferation in 2D and 3D culture systems. Lapatinib is able to promote heterodimerisation between the kinase domains of full-length HER2 and HER3 in cells. However, this dimer interface is different from the canonical active EGFR-family dimer, and it is necessary for the lapatinib/NRG combinatorial proliferative phenotype. Both the lapatinib-induced heterodimer and the cooperative proliferation effects depend strongly on the ability for the pseudokinase HER3 to bind ATP. Consistent with the model, occupying the pseudokinase HER3 with the Src/Abl inhibitor bosutinib stabilises the pseudokinase domain to the extent that it actually promotes HER2-HER3 heterodimerisation and downstream proliferation.

## Results

### Lapatinib-NRG co-treatment shows a synergistic effect on proliferation, dependent on HER3 ATP binding

The sensitivity of a variety of oncogene-addicted cell lines to small molecule kinase inhibitors can be counter-acted by the addition of growth factors (*Wilson et al., 2012*). This includes the case of lapatinib-treated HER2$^+$ breast cancer cell lines, where NRG is seen to mediate a rescue of drug toxicity (*Novotny et al., 2016*; *Wilson et al., 2012*). Using different experimental procedures, we have investigated further these competing effects of lapatinib and NRG on the proliferative behaviour of HER2$^+$ breast cancer cells.

In SKBR3, BT474, AU565, and HCC1419 cells treated with a range of lapatinib concentrations for 72 hr, the addition of 10 nM NRG rescues the drug-induced cytotoxicity except at very high drug concentrations (*Figure 1a*, *Figure 1—figure supplement 1a–c*).

Interestingly, in the case of the SKBR3, BT474 and AU565 cell lines, low concentrations of lapatinib (~40–400 nM) are able to enhance proliferation in conjunction with 10 nM NRG by 25–30% compared to growth factor alone (*Figure 1a*, *Figure 1—figure supplement 1a–b*). A partial response of this cooperative phenotype is observed in ZR75 and HCC1419 cells (*Figure 1—figure supplement 1c–d*). This phenotype in SKBR3 cells, while observed previously, has gone unremarked (*Novotny et al., 2016*; *Wilson et al., 2012*). We corroborated our results with a cell counting assay, in which SKBR3 cells were treated for 72 hr with 250 nM lapatinib or vehicle ±10 nM NRG (*Figure 1b*). The emergent effect of lapatinib plus NRG depends on lapatinib sensitivity. Two breast cancer cell lines with low lapatinib sensitivity, MCF7 and HCC1569, show low inhibitor-growth factor cooperation (*Figure 1—figure supplement 1e–f*). The growth phenotype in ZR75 may be partially explained by its HER4 expression, considering that NRG is also a ligand for HER4 (*Figure 1—figure supplement 1g*).

Although HER3 has been shown to bind lapatinib *in vitro* with very low affinity (Kd = 5.5 μM) (*Davis et al., 2011*), the synergistic behaviour between lapatinib and NRG occurs in cells at a ~ 50 x lower dose than the *in vitro* Kd, indicating that any binding of lapatinib to HER3 would likely be minor under these conditions. Using a thermal shift assay (TSA), which measures a shift in the thermal stability of a protein after ligand/inhibitor binding *in vitro*, we also show that lapatinib does not strongly bind HER3 as compared to ATP and a panel of other inhibitors (*Figure 2a*, see further below).

While EGF treatment rescued SKBR3 cells from the effects of low-concentration lapatinib treatment, synergistic growth effects such as those observed with lapatinib-NRG co-treatment were not observed for lapatinib-EGF co-treated SKBR3 or BT474 cells (*Figure 1—figure supplement 1h–i*). Although NRG is also a growth factor ligand for HER4, protein levels of HER4 in SKBR3 cells are very low (*Figure 1—figure supplement 1g*). Additionally, lapatinib is a strong inhibitor of both EGFR and

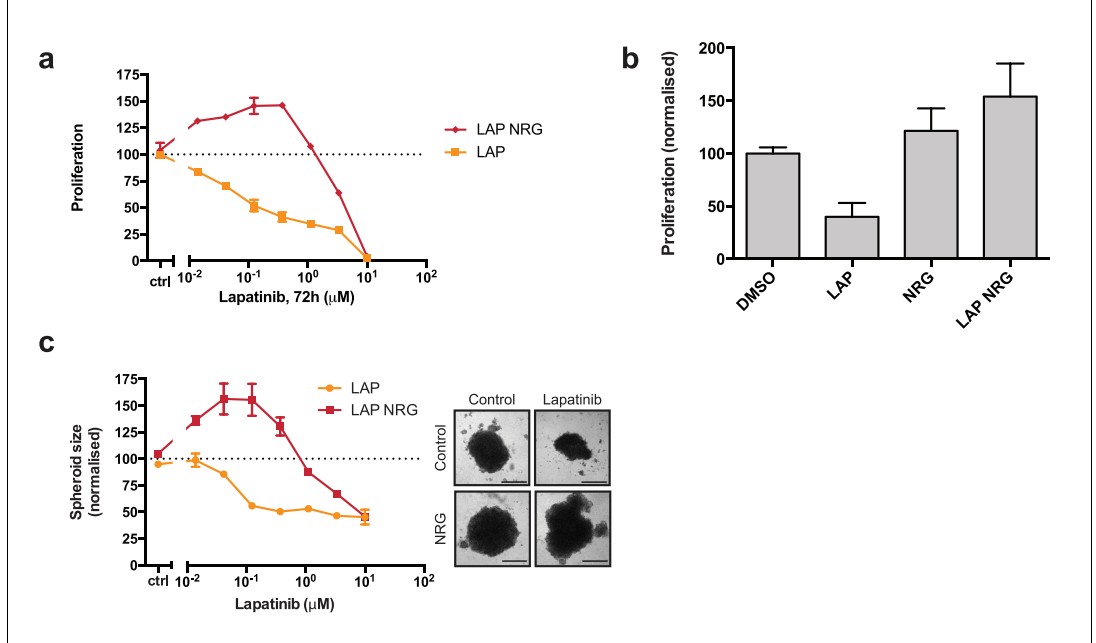

**Figure 1.** Lapatinib and NRG have synergistic effects on SKBR3 growth in 2D and 3D culture systems. (**a**) CellTiter-Glo proliferation assay of SKBR3 cells after treatment for 72 hr with a range of lapatinib concentrations ± 10 nM NRG. (**b**) Cell counting assay of SKBR3 cells treated for 72 hr with DMSO or 250 nM lapatinib ±10 nM NRG, before quantification of cell number on a Vi-CELL counter. (**c**) Quantification of SKBR3 3D spheroid area after 8 days of treatment with a range of lapatinib concentrations ± 10 nM NRG, with representative bright field micrographs. Scale bars 0.5 mm. All proliferation data represented as mean ±SEM of three independent experiments each performed in quadruplicate. Corresponding data and statistics available as *Figure 1—source data 1*.

DOI: https://doi.org/10.7554/eLife.32271.003

The following source data and figure supplements are available for figure 1:

**Source data 1.** Numerical data and statistics relating to *Figure 1*.
DOI: https://doi.org/10.7554/eLife.32271.008

**Figure supplement 1.** Effects of lapatinib and NRG on breast cancer cell proliferation.
DOI: https://doi.org/10.7554/eLife.32271.004

**Figure supplement 1—source data 1.** Numerical data and statistics relating to *Figure 1—figure supplement 1*.
DOI: https://doi.org/10.7554/eLife.32271.005

**Figure supplement 2.** The irreversible inhibitor neratinib does not show synergistic growth under ligand co-treatment conditions.
DOI: https://doi.org/10.7554/eLife.32271.006

**Figure supplement 2—source data 1.** Numerical data and statistics relating to *Figure 1—figure supplement 1*.
DOI: https://doi.org/10.7554/eLife.32271.007

HER4 (*Davis et al., 2011*). Taken together, these data seem to exclude a significant role for EGFR and HER4 in the synergistic growth observed for lapatinib-NRG co-treatment. Moreover, transient knockdown of HER3 with two different siRNA oligonucleotides shows a modest, but consistent reduction in the proliferative effect of ligand-inhibitor co-treatment, implicating HER3 as the relevant growth factor-binding receptor for this NRG response (*Figure 1—figure supplement 1j*).

The proliferative effects of lapatinib and NRG on SKBR3 cells were also observed in 3D spheroid cultures. As seen in 2D culture systems, in 3D spheroid culture the addition of NRG to lapatinib-treated cells rescues SKBR3 cells from lapatinib-induced cytotoxicity/cytostasis (*Figure 1c*, *Figure 1—figure supplement 1k–l*). Lapatinib and NRG share a cooperative effect on the induction of proliferation in 3D spheroid cultures, where spheroid size is greater for inhibitor-ligand co-treatment conditions than for those treated with growth factor alone.

The irreversible inhibitor neratinib binds the same inactive conformation as lapatinib and with similar binding affinity (*Davis et al., 2011*). However, neratinib is an irreversible inhibitor and forms a covalent bond with HER2$^{C805}$, a residue conserved in EGFR and HER4 but not HER3. Neratinib-NRG co-treatment did not show the synergistic proliferative phenotype observed with lapatinib-NRG, in

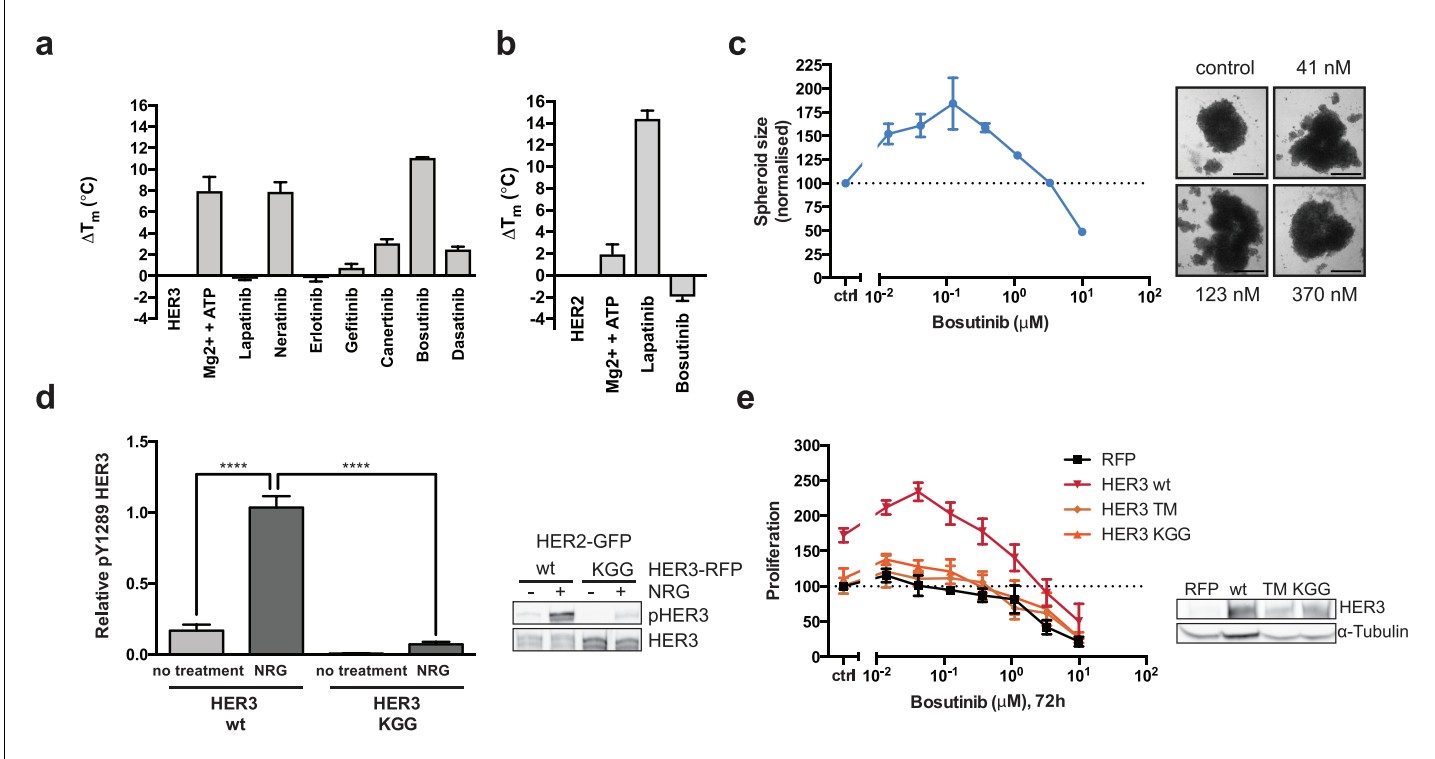

**Figure 2.** HER3 ATP-binding pocket occupation is necessary and sufficient to drive SKBR3 cell growth. (a) *In vitro* TSA binding assay of HER3 with selected kinase inhibitors. (b) *In vitro* TSA binding assay of HER2 with lapatinib and bosutinib. (c) Quantification of spheroid size after eight days of treatment with a titration of bosutinib with representative bright field micrographs of SKBR3 cell spheroids after eight days of bosutinib treatment. Scale bars signify 0.5 mm. (d) Transient co-transfection of MCF7 cells with HER2wt-GFP and HER3wt-RFP or HER3KGG-RFP. Cells were serum starved for one hour, followed by 10 nM NRG or vehicle for ten minutes. HER3 phosphorylation on Y1289 was measured by Western blot and analysed by densitometry relative to total HER3. (e) SKBR3 cells were transfected with RFP empty vector, HER3wt-RFP, HER3T787M-RFP or HER3KGG-RFP. 72 hr of bosutinib treatment was initiated 24 hr post-transfection. Proliferation was measured using CellTiter-Glo. TSA data represented as mean ±SEM of (a) two independent experiments each performed quadruplicate, or (b) three independent experiments each performed in at least quadruplicate. Proliferation data represented as mean ±SEM of three independent experiments each performed in at least triplicate. Western blot data shown as mean ±SD for three independent experiments. Western blot quantifications analysed by one-way ANOVA. ****p≤0.0001 Corresponding data and statistics available as *Figure 2—source data 1*.

DOI: https://doi.org/10.7554/eLife.32271.009

The following source data and figure supplements are available for figure 2:

**Source data 1.** Numerical data and statistics relating to *Figure 2*.
DOI: https://doi.org/10.7554/eLife.32271.013

**Figure supplement 1.** The effects of HER3 ATP-binding pocket occupation on drug-induced cell proliferation.
DOI: https://doi.org/10.7554/eLife.32271.010

**Figure supplement 1—source data 1.** Numerical data and statistics relating to *Figure 2-figure supplement 1*.
DOI: https://doi.org/10.7554/eLife.32271.011

**Figure supplement 2.** Cell surface expression of HER3 mutants.
DOI: https://doi.org/10.7554/eLife.32271.012

either a cell counting assay, or in 3D spheroid formation (*Figure 1—figure supplement 2a–d*). Similarly, the induction of HER2 and HER3 phosphorylation seen in western blot analysis of lapatinib-NRG co-treated 3D spheroids was absent in neratinib-NRG co-treatment (*Figure 1—figure supplement 1l*, *Figure 1—figure supplement 2d*). This indicates that the proliferative phenotype observed for lapatinib is likely to necessitate a dynamic, reversible inhibitor binding.

Collectively, the data from both 2D and 3D cultures show that there is a counterintuitive synergy between the HER2 inhibitor lapatinib and the HER3 ligand NRG in driving the proliferation of SKBR3 cells. This prompted us to examine the potential for novel allosteric regulation of HER2-HER3 heterotypic interactions by both ligand and inhibitors.

## HER3 nucleotide pocket occupation is of structural importance

To study the effects of ATP binding on HER3 function, we aimed to both stabilise and destabilise the pseudokinase nucleotide-binding pocket. This would allow us to investigate the importance of the structural role that nucleotide binding pocket occupation has been shown to play in several (pseudo)kinases.

To separate the structural and trace catalytic roles that ATP-binding could fulfill in HER3, we used the ATP-competitive Src/Abl inhibitor bosutinib, which has been shown to bind strongly to HER3 but not to other EGFR family members (*Levinson and Boxer, 2014*; *Davis et al., 2011*). We compared bosutinib to a small panel of EGFR family inhibitors as well as an additional Src inhibitor, dasatinib, and measured HER3 thermal stability by TSA (*Figure 2a*, *Figure 2—figure supplement 1a*). In line with previous observations, we confirmed that HER3 strongly binds bosutinib. Significantly, lapatinib was not able to provide a noticeable thermal shift, which corresponds to previously published results indicating HER3 does not bind lapatinib with high affinity (*Davis et al., 2011*). While lapatinib was able to confer strongly increased thermal stability to HER2, bosutinib was not (*Figure 2b*). This is in line with previously published data that indicates HER2 is not a strong bosutinib binder (*Davis et al., 2011*).

We hypothesised that bosutinib might be able to aid proliferation in a cellular context by stabilising the nucleotide binding pocket of HER3 and helping sustain dimer formation, analogous to vemurafenib-bound behaviour of BRAF. In a 2D proliferation assay, SKBR3 cells treated with bosutinib over 72 hr show a dose dependent induction of proliferation without additional NRG stimulation (*Figure 2—figure supplement 1b*). This proliferative effect is sustained in eight-day treatments in 3D spheroid cultures (*Figure 2c*, *Figure 2—figure supplement 1d,e*). The ability of bosutinib to induce SKBR3 cell proliferation appears to be an EGFR-family mediated event, as lapatinib treatment can curtail its effects in a dose-dependent manner (*Figure 2—figure supplement 1e*).

In order to destabilise the HER3 nucleotide binding pocket we made the triple mutant HER3$^{KGG}$. HER3$^{K742}$ was mutated to methionine to hinder ATP α-phosphate coordination, which by itself has been shown to reduce HER3 mant-ATP binding affinity (*Shi et al., 2010*). To obstruct ATP binding further, double aspartates were introduced in the glycine-rich loop (HER3$^{G716D/G718D}$) to mimic the pseudokinase-specific aspartate residue observed in the glycine-rich loop of VRK3 (*Scheeff et al., 2009*), adding a negative charge in the area where the ATP phosphates would normally sit. Introduction of this ATP-binding deficient HER3$^{KGG}$ mutant into MCF7 cells shows abrogation of ligand-induced trans-phosphorylation of HER3 by HER2 (*Figure 2d*). SKBR3 cells ectopically expressing HER3$^{wt}$ or HER3$^{KGG}$ show a differential proliferative behaviour upon lapatinib ±NRG treatment. This indicates a critical role for HER3 ATP binding in order to sustain inhibitor-growth factor cooperative proliferation (*Figure 2—figure supplement 1f*).

The bosutinib binding of HER3$^{wt}$, HER3$^{KGG}$, and the proposed drug de-sensitised HER3$^{T787M}$ (*Levinson and Boxer, 2014*; *Dong et al., 2017*), was investigated using an in-cell thermal shift assay (CETSA) (*Jafari et al., 2014*; *Reinhard et al., 2015*). Where wild type HER3 showed increased thermal stability in cells in the presence of 50 nM bosutinib, HER3$^{KGG}$ did not (*Figure 2—figure supplement 1g*). Ectopic expression of wild type HER3, but not HER3$^{KGG}$ or HER3$^{T787M}$, enhances bosutinib-mediated proliferation, indicating this behaviour is driven by bosutinib binding to HER3 directly (*Figure 2e*). Both HER3$^{KGG}$ and HER3$^{T787M}$ showed normal localization to the plasma membrane, as measured by flow cytometry, indicating that these mutations did not compromise the receptor and its traffic to the plasma membrane (*Figure 2—figure supplement 2*).

The HER3$^{KGG}$ and bosutinib results indicate that nucleotide pocket occupation in HER3 is essential for its ability to sustain a proliferative signalling pathway under distinct circumstances: in the acute response to growth factor, in promoting ligand-inhibitor cooperative proliferation and even after treatment with a HER3-binding inhibitor. This indicates a critical structural role for HER3 ATP-binding pocket occupation in its ability to sustain heterointeractions and proliferation. Considering the proliferative effects observed with the HER3-binding inhibitor bosutinib, our results also suggest that any residual transferase activity HER3 retains does not appear to be important in these responses *in vivo* unless we invoke a hit-and-run mechanism of action for bosutinib on HER3 which would seem unlikely.

## Lapatinib binding induces HER2-HER3 heterodimerisation

The stability conferred to a protein kinase by small molecule inhibitor binding has been shown to play an important role in the promotion of protein–protein interactions. We investigated the potential role of lapatinib to similarly promote HER2-HER3 heterodimerisation by stabilising particular protein conformations in HER2 using a FRET-FLIM approach. We measured drug-induced heterodimerisation of HER2 and HER3, as we have done previously in the case of drug-induced dimerisation of the EGF receptor (*Bublil et al., 2010*; *Coban et al., 2015*).

At endogenous protein levels in SKBR3 cells, we observe lapatinib-driven HER2-HER3 heterodimerisation to levels similar to those seen with NRG (*Figure 3a*). Interestingly, the lapatinib-induced dimerisation occurs in the absence of exogenously added NRG, indicating a HER2-HER3 dimer that is driven primarily through intracellular domain interactions. MCF7 cells, which express low levels of endogenous HER2 and HER3 compared to SKBR3, also display lapatinib-induced heterodimerisation of ectopically expressed GFP-HER2$^{wt}$ and HA-HER3$^{wt}$ (*Figure 3b*).

As discussed above, occupation of the nucleotide binding pocket in HER3 is of importance for its ability to sustain proliferation. This is also reflected in the case of lapatinib-induced heterodimer formation, where the introduction of the nucleotide pocket compromised HER3$^{KGG}$ mutant strongly disrupts inhibitor-promoted heterodimerisation (*Figure 3c*). In line with the proliferative effects described above, bosutinib was also able to directly promote heterodimerisation between HER2 and HER3 (*Figure 3d*).

Using stochastic optical reconstruction microscopy (STORM), we analysed receptor clustering in SKBR3 cells. Treatment with either NRG, lapatinib, or bosutinib showed a shift in cluster population size compared to control, implying the formation of higher-order oligomers rather than dimers (*Figure 3e,f*). The exact HER2-HER3 stoichiometry in these drug-treated oligomers remains elusive, because these experimental conditions allowed us to count only cluster size for either HER2 or HER3, not both at the same time. Therefore, it is expected that the observed HER3 clusters also contain uncounted HER2 receptors, and vice versa, as evident in the FRET-FLIM data.

## Disruption of the active HER2-HER3 interface

The active signalling dimer in the EGFR family adopts an asymmetric orientation, in which there is a distinct division of labour in the activator-receiver pairing. One kinase (the activator kinase) does not phosphorylate substrates, but binds in a way that helps its heterodimerisation partner (the receiver kinase) in adopting an active conformation. The receiver kinase is then capable of substrate phosphorylation. Originally described for EGFR homodimerisation, and similar to the cyclin/CDK binding mode (*Jeffrey et al., 1995*; *Zhang et al., 2006*), this canonical active dimerisation interface has been reported across the EGFR family including the heterodimerisation of HER3, which can only perform the activator role (*Jura et al., 2009b*; *Littlefield et al., 2014*; *van Lengerich et al., 2017*). Mutations that disrupt this active interface in both the activator and receiver partner kinases are well-documented and are schematically highlighted (*Figure 4a*, *Figure 4—video 1*).

In the case of the active, activator/receiver interface, HER3 buttresses the inward orientation of the HER2 α-C helix, leaving no space for the HER2 α-C helix to adopt the 'out' orientation characteristic of the inactive conformation. We modelled the potential effects of HER2 α-C helix positioning on lapatinib binding to test whether canonical activator/receiver orientation (in which the HER2 α-C helix is pushed inwards) would give sufficient space to still accommodate lapatinib. Our modelling showed that, for a HER2 α-C helix in the active, 'in' position, lapatinib binding results in a potential steric clash with HER2$^{E770/M774}$ (*Figure 4—figure supplement 1a,b*). A general decrease of the nucleotide binding pocket volume from 756 Å$^3$ to 232 Å$^3$ (calculated using SURFNET v1.5 (*Laskowski, 1995*)) supports these predictions.

To further test whether the lapatinib-induced HER2-HER3 is adopting the canonical activator/receiver orientation, we used FRET-FLIM to investigate lapatinib-induced dimer formation. The I714Q mutation in HER2, which renders the receptor receiver-impaired, disrupted the lapatinib-driven HER2-HER3 association, indicating it is retained in the lapatinib-induced dimer interface (*Figure 4b*). However, the reciprocal activator-impaired mutation in HER3 (HER3$^{V945R}$) did not disrupt lapatinib-mediated heterodimerisation, although it efficiently suppressed the canonical active dimer after ligand-induced heterodimerisation (*Figure 4c*).

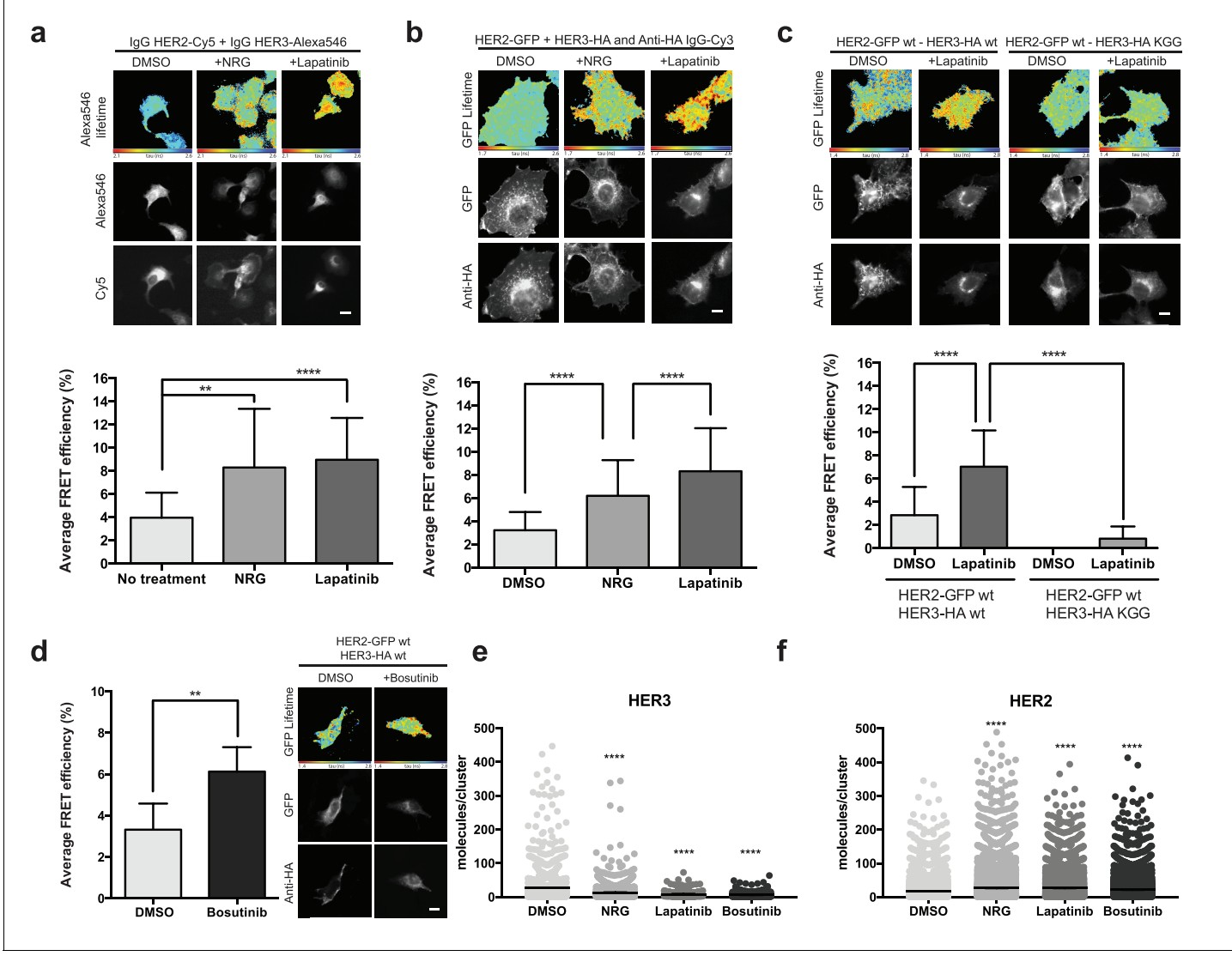

**Figure 3.** Inhibitor-induced HER2-HER3 heterotypic interactions. (**a**) FRET-FLIM analysis of endogenous HER2-HER3 association in SKBR3 cells, serum starved for 1 hr, and stimulated with 6.7 nM NRG for 15 min, or inhibited with lapatinib (10 μM) for 1 hr, prior to fixation and staining with IgG anti-HER2-Cy5 and IgG anti-HER3-Alexa546 overnight, at 4°C. (**b**) MCF7 cells were transfected with vectors encoding HER2$^{wt}$-GFP and HER3$^{wt}$-HA. Cells were incubated as in (**a**) and stained with anti-HA antibody conjugated to Alexa-546 (controls treated with vehicle). (**c**) MCF7 cells were transfected with vectors encoding HER2$^{wt}$-GFP and HER3$^{wt}$-HA or HER3$^{KGG}$-HA. Cells treated with lapatinib (10 μM) for 1 hr, prior to fixation and staining with anti-HA antibody conjugated to Alexa-546. (**d**) SKBR3 cells were treated with bosutinib (50 nM, 1 hr), and stained as in (**b**). (**e, f**) Molecules/cluster measurements from STORM data taken of SKBR3 cells labelled with HER2Affibody-Alexa488 and HER3Affibody-Alexa647 or NRG-Alexa647 ±14 nM lapatinib or 41 nM bosutinib. Cumulative FRET-FLIM histograms show average FRET efficiency from three independent experiments. **p≤0.01; ****p≤0.0001 Scale bars 5 μm. Clustering data represent mean combination of two independent experiments with each measuring >1000 clusters. Clustering data presented as mean with 95% CI. Corresponding data and statistics available as *Figure 3—source data 1*.

DOI: https://doi.org/10.7554/eLife.32271.014

The following source data is available for figure 3:

**Source data 1.** Numerical data and statistics relating to *Figure 3*.
DOI: https://doi.org/10.7554/eLife.32271.015

It is surmised that the inhibitor binding is able to robustly induce a heterodimer between HER2 and HER3, which is distinct from the canonical active heterodimer induced after growth factor stimulation. The orientation of this non-canonical lapatinib-driven heterodimer retains HER2$^{I714}$ in the

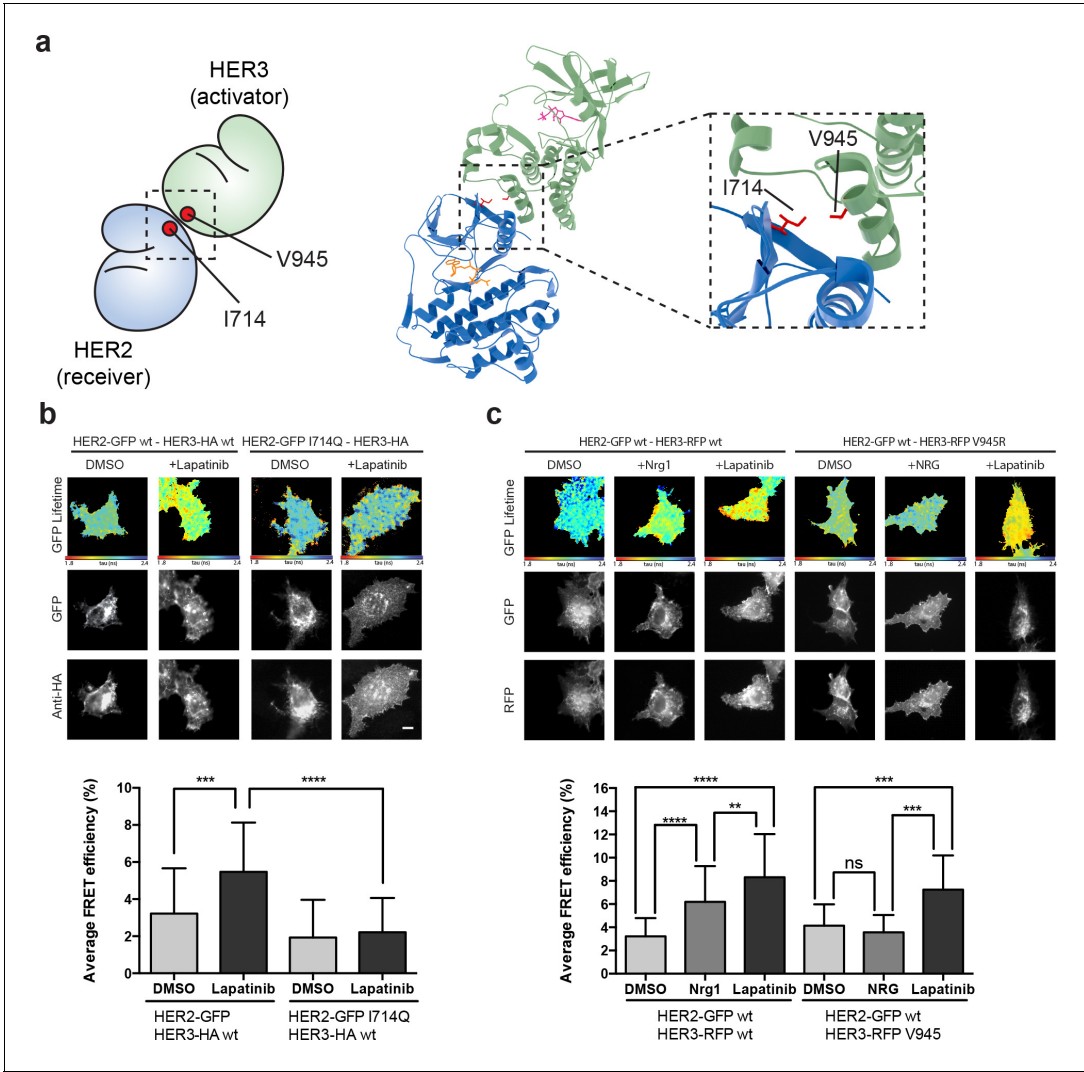

**Figure 4.** The lapatinib-induced HER2-HER3 dimer is distinct from the active, asymmetric HER2-HER3 dimer orientation. (**a**) Schematic representation and molecular model of HER2-HER3 active, asymmetric kinase domain dimer orientation. Inset denotes interaction interface. (**b**) MCF7 cells were transfected with vectors encoding HER2$^{wt}$-GFP or HER2$^{I714Q}$-GFP and HER3$^{wt}$-HA. Cells were treated as described in *Figure 3* and HER2-HER3 association was measured by FRET-FLIM. (**c**) MCF7 cells were transfected with vectors encoding HER2-GFP and HER3$^{wt}$-RFP or HER3$^{V945R}$-RFP. Cells were incubated as described above, and treated with DMSO, lapatinib or NRG prior to fixation. Data represents mean ±SEM. *p≤0.05; **p≤0.01; ***p≤0.001; ****p≤0.0001 by one-way ANOVA. Scale bars represent 5 μm. Corresponding data and statistics available as *Figure 4—source data 1*. Molecular model for the interaction in (a) available as *Figure 4—source data 2*.
DOI: https://doi.org/10.7554/eLife.32271.016

The following video, source data, and figure supplements are available for figure 4:

**Source data 1.** Numerical data and statistics relating to *Figure 4*.
DOI: https://doi.org/10.7554/eLife.32271.020

**Source data 2.** PDB structure file of molecular interaction model in *Figure 4a*.
DOI: https://doi.org/10.7554/eLife.32271.021

**Figure supplement 1.** Model of lapatinib binding in HER2 inactive and active conformations shows a potential steric clash.
DOI: https://doi.org/10.7554/eLife.32271.017

**Figure supplement 1—source data 1.** PDB structure file of inhibitor docking model in *Figure 4—figure supplement 1a*.
DOI: https://doi.org/10.7554/eLife.32271.018

**Figure supplement 1—source data 2.** PDB structure file of inhibitor docking model in *Figure 4—figure supplement 1b*.
DOI: https://doi.org/10.7554/eLife.32271.019

**Figure 4—video 1.** Interface view of the molecular model of an active HER2-HER3 heterodimer, with HER2$^{I714}$ and HER3$^{V945}$ highlighted.
DOI: https://doi.org/10.7554/eLife.32271.022

dimer interface, giving us a starting point for in silico molecular modelling to investigate potential dimer conformations distinct from the well-described active dimer.

## Lapatinib drives a novel HER2-HER3 heterodimerisation interface

In the case of type II kinase inhibitors such as lapatinib, the inhibitor stabilises an inactive conformation of the kinase domain, where the α-C helix is tilted outwards. As HER3 lacks the conserved glutamate residue in the α-C helix, HER3$^{K742}$ is unable to form the salt bridge normally observed in active kinase domain structures (*Huse and Kuriyan, 2002*). The HER3 ATP-bound conformation therefore does not show a classical active conformation with the α-C helix tilted inward (*Jura et al., 2009b*; *Shi et al., 2010*), but instead resembles the inactive conformation seen in kinases bound to type II inhibitors such as lapatinib. Because lapatinib-bound HER2 and ATP-bound HER3 adopt similar conformations, there is a possibility that the lapatinib-induced, inactive dimer is oriented symmetrically.

In the crystal lattices of EGFR and HER3 kinase domains, two different symmetrical interaction interfaces have been observed (*Jura et al., 2009a*; *Jura et al., 2009b*). We used molecular modelling to investigate the potential for HER3 and lapatinib-bound HER2 to adopt either of these conformers (*Figure 5a–b*, *Figure 5—figure supplement 1a–b*). HER2$^{I714}$ is present in the interaction interface of both the EGFR-like, staggered orientation, as well as in the head-to-head, HER3-like orientation. This falls in line with the FRET-FLIM data in *Figure 4* that suggests the retained presence of the HER2$^{I714}$ residue in the lapatinib-induced dimer interface.

On the basis of these models, we designed pairs of mutations in HER2 that would exclusively disrupt one of the potential heterodimer orientations (*Figure 5—figure supplement 1*, *Figure 5— video 1* and *2*). For the EGFR-like, staggered dimer we substituted two hydrophobic residues on HER2 with two positively charged residues, HER2$^{I748R/V750R}$, which should lead to repulsion from the positively charged residues, K998 and K999, lying on the HER3 side of the interface.

Likewise, for the HER3-like, head-to-head dimer, we predicted that the HER2$^{N764R/K765F}$ mutant would disrupt the dimerisation interface. The substitution of an asparagine residue (HER2$^{N764}$) with a positively charged arginine should lead to repulsion from a positively charged HER3 residue (HER3$^{R702}$), lying within a radius of 4 Å and opposite to HER2$^{N764}$, therefore causing severe disruption of the HER3-like dimer interface. Furthermore, the substitution of a lysine residue (HER2$^{K765}$) with a bulky, hydrophobic residue such as phenylalanine should generate clashes at this HER2-HER3 interface.

These dimer interface mutants were introduced into our FRET-FLIM assay for investigation of the lapatinib-induced heterodimerisation conformer (*Figure 5c*). The HER2$^{N764R/K765F}$ mutant disrupted heterodimerisation upon lapatinib binding, whereas HER2$^{I748R/V750R}$ showed no difference in heterodimer formation. These mutational FRET/FLIM data is consistent with our model that the lapatinib-induced HER2-HER3 heterodimer adopts a symmetrical, head-to-head orientation, similar to the one observed in the HER3 kinase domain crystal lattice (*Jura et al., 2009b*) (*Figure 5b*).

## Head-to-head HER2-HER3 dimerisation is required for inhibitor-induced proliferation

Having presented modelling and FRET/FLIM data consistent with an orientation of the lapatinib-induced HER2-HER3 dimer being distinct from the active activator/receiver dimer interface, we sought to identify which type of HER2-HER3 interaction caused the NRG-lapatinib co-stimulatory growth observed in 2D proliferation assays.

In these assays, we did not ectopically introduce the HER2$^{N764R/K765F}$ mutant because, firstly, it might also disrupt the active, asymmetrical HER2-HER3 heterodimer interface and secondly, SKBR3 cells have vast numbers of endogenous HER2 receptors that would hinder analysis of the behaviour of ectopically expressed HER2$^{N764R/K765F}$. Instead we identified HER3$^{L700F}$ as the reciprocal mutant to HER2$^{N764R/K765F}$ (*Figure 6a*, *Figure 6—video 1*). We introduced HER3$^{L700F}$ into SKBR3 cells to investigate the role of the head-to-head, symmetric dimer interface in the lapatinib-NRG synergistic proliferation described above. While the HER3$^{V945R}$ active dimer mutant did not disrupt drug-growth factor cooperative proliferation, the HER3$^{L700F}$ mutant did (*Figure 6d–e*). Both HER3$^{L700F}$ and HER3$^{V945R}$ were expressed on the cell surface, as measured by flow cytometry (*Figure 2—figure supplement 2*). Combined, this indicates that the inhibitor-induced heterodimer of HER2 and HER3 is consistent with a head-to-head, symmetrical conformation, and it plays an important role in the

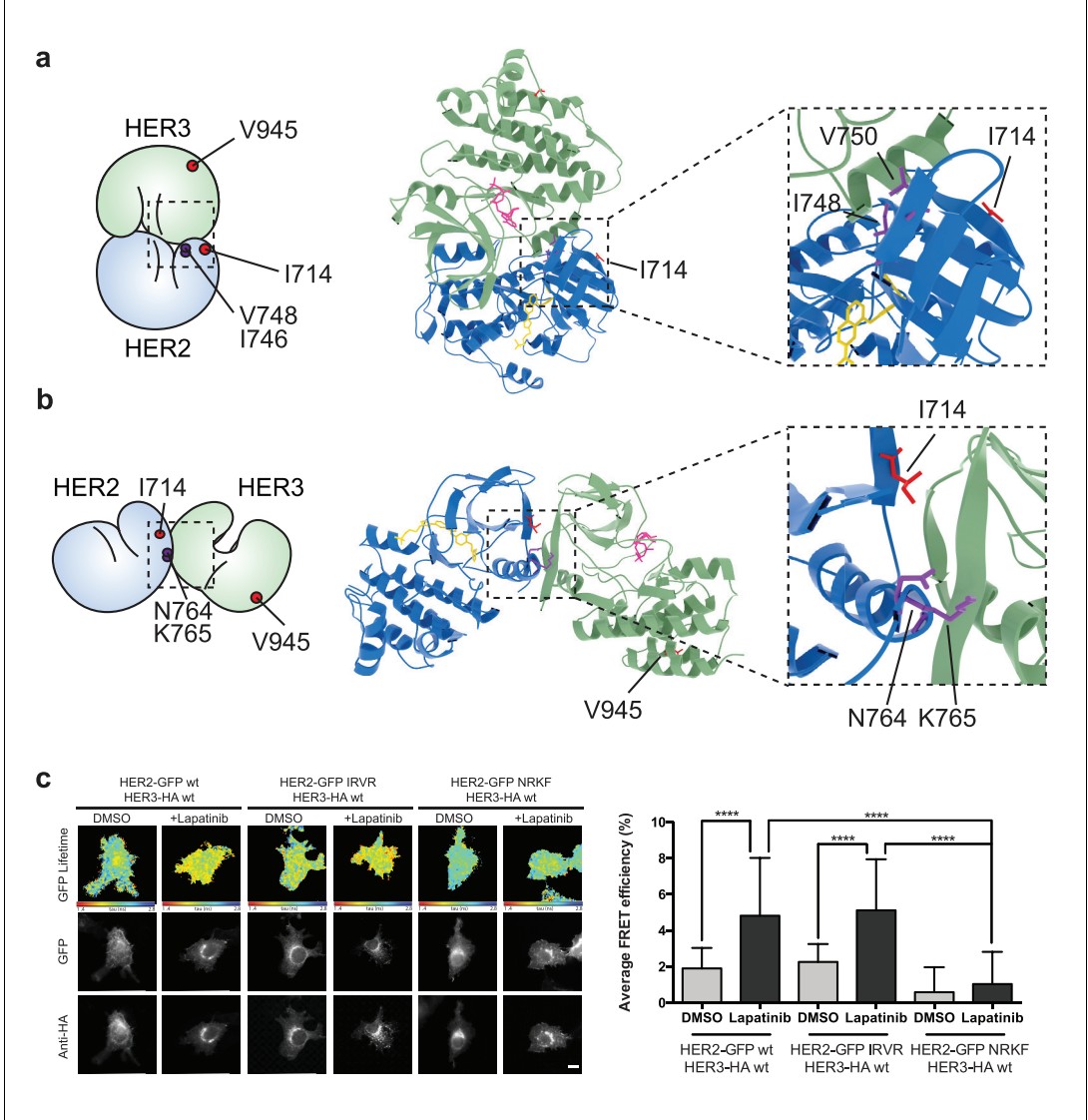

**Figure 5.** The lapatinib-induced HER2-HER3 dimer is in a symmetric orientation. (**a**) Lapatinib-bound HER2 (blue, lapatinib in yellow) and ATP analogue-bound HER3 (green, AMP-PNP in pink) were modelled in an EGFR-like symmetric dimer orientation (*Jura et al., 2009a*). Inset highlights the interaction interface. The schematic representation shows active dimer interface residues HER2[I714] and HER3[V945], as well as the two residues in HER2 unique to this interface for further mutational analysis. (**b**) Lapatinib-bound HER2 in the HER3-like head-to-head symmetric dimer orientation (*Jura et al., 2009b*). Dimer-specific residues are highlighted in the schematic. (**c**) MCF7 cells were transfected with vectors encoding HER2[wt]-GFP, HER2[N764R/K765F]-GFP or HER2[I748R/V750R]-GFP and HER3[wt]-HA. Cells were incubated for 24 hr, and inhibited with 10 μM lapatinib for 1 hr, prior to fixation and staining with anti-HA antibody conjugated to Alexa-546. Data represented as mean ±SEM. ****p≤0.0001, as analysed by one-way ANOVA. Scale bars 5 μm corresponding data and statistics available as *Figure 5—source data 1*. Molecular model for the interactions in (**a**) and (**b**) available as *Figure 5—source data 2* and *3*. Residues marking the dimer interface of the lapatinib-induced HER2-HER3 heterodimer, in either the EGFR-like or HER3-like modelled conformations, including the per-residue solvent accessible surface area (in Å$^2$) are available as *Figure 5—source data 4*.
DOI: https://doi.org/10.7554/eLife.32271.023

The following video, source data, and figure supplement are available for figure 5:

**Source data 1.** Numerical data and statistics relating to *Figure 5*.
DOI: https://doi.org/10.7554/eLife.32271.025

**Source data 2.** PDB structure file of molecular interaction model in *Figure 5a*.
DOI: https://doi.org/10.7554/eLife.32271.026

**Source data 3.** PDB structure file of molecular interaction model in *Figure 5b*.
DOI: https://doi.org/10.7554/eLife.32271.027

**Source data 4.** Table with modelled interface residues, including the per-residue solvent-accessible surface area in Å$^2$.

*Figure 5 continued on next page*

*Figure 5 continued*

DOI: https://doi.org/10.7554/eLife.32271.028

**Figure supplement 1.** Molecular models of potential orientations of the lapatinib-induced HER2-HER3 dimer.

DOI: https://doi.org/10.7554/eLife.32271.024

**Figure 5—video 1.** Interface view of the molecular model of a lapatinib-induced HER2-HER3 heterodimer in the EGFR-like conformation, with HER2$^{I714}$ and HER3$^{V945}$ highlighted, as well as model-specific interface residues HER2$^{I748/V750}$.

DOI: https://doi.org/10.7554/eLife.32271.029

**Figure 5—video 2.** Interface highlight of the molecular model of a lapatinib-induced HER2-HER3 heterodimer in the HER3-like conformation, with HER2$^{I714}$ and HER3$^{V945}$ highlighted, as well as model-specific interface residues HER2$^{N764/K765}$.

DOI: https://doi.org/10.7554/eLife.32271.030

synergistic proliferative effects of lapatinib and NRG. Although this conformation has been described from the HER3 kinase domain crystal lattice (*Jura et al., 2009b*), to our knowledge it is the first time a functional role has been ascribed to heterodimers consistent with this interface in cells.

## Discussion

The conformational dynamics of HER2-HER3 heterodimerisation are an important consideration for evaluating existing and future targeted therapy intervention strategies against HER2$^+$ breast cancer and other HER family driven cancers. Here, we show that the HER2 inhibitor lapatinib is paradoxically able to promote proliferative behaviour in HER2$^+$ breast cancer cells when administered in the presence of the HER3 ligand NRG. The synergy between growth factor and inhibitor requires an intricate, multi-step cascade of conformational events.

Lapatinib itself is able to promote heterodimerisation between the kinase domains of HER2 and HER3, stabilising an orientation consistent with a symmetric, head-to-head kinase domain heterodimer that is distinct from the canonical, asymmetric, head-to-tail active kinase domain orientation that occurs throughout the EGFR family. An analogous interface has previously been observed in the HER3 kinase domain crystal lattice (*Jura et al., 2009b*); here, we have provided modelling and cellular evidence of a heterodimer with an interface consistent to the one observed in the HER3 kinase domain crystal lattice. Sequestering HER2 and HER3 in these inactive, lapatinib-bound heterodimers was of benefit to NRG-mediated proliferative signalling. Our results, in which inhibitor binding drives dimer formation that boosts signalling and proliferation, shows some parallels with the inhibitor-induced signalling phenotypes in the RAF-family (*Eyers et al., 1998*; *Hatzivassiliou et al., 2010*; *McKay et al., 2011*; *Poulikakos et al., 2010*; *Thevakumaran et al., 2015*)

While the FRET-FLIM analysis of the lapatinib-induced dimerisation was not able to differentiate between heterodimers or higher order oligomers, our clustering data shows that lapatinib is likely to induce higher order oligomers. Because of the modelled symmetrical nature of these lapatinib-induced dimers, in which both lapatinib-bound HER2 and HER3 would be conformationally available as 'activator' receptors for additional oligomerization partners, it is not inconceivable they may act as nucleation points for larger oligomeric signalling platforms. Such signalling arrays, in which mutual cooperativity increases signaling output, have been proposed for EGFR oligomers (*Huang et al., 2016*).

The addition of ligand potentially causes rearrangements within these platforms through the ligand-induced conformational ballet of multi-level interactions between the various extracellular and intracellular domains of EGFR family receptors (reviewed in (*Lemmon et al., 2014*)). The formation of lapatinib-induced oligomeric platforms may facilitate a transition into active signalling heterodimers upon ligand binding, due to the availability of dimerisation partners in immediate proximity within these drug-induced oligomer platforms.

Both the lapatinib-induced HER2-HER3 heterodimerisation and the downstream lapatinib-NRG synergistic effects on proliferation depended on the ability of HER3 to bind ATP. Although usually classified as a pseudokinase, HER3 has been shown to retain a measure of autophosphorylation activity (not transphosphorylation) under specific circumstances (*Shi et al., 2010*). We show HER2-HER3 heterodimerisation and downstream proliferative effects can be elicited by the addition of the HER3-binding inhibitor bosutinib, indicating that nucleotide binding pocket occupation performs a

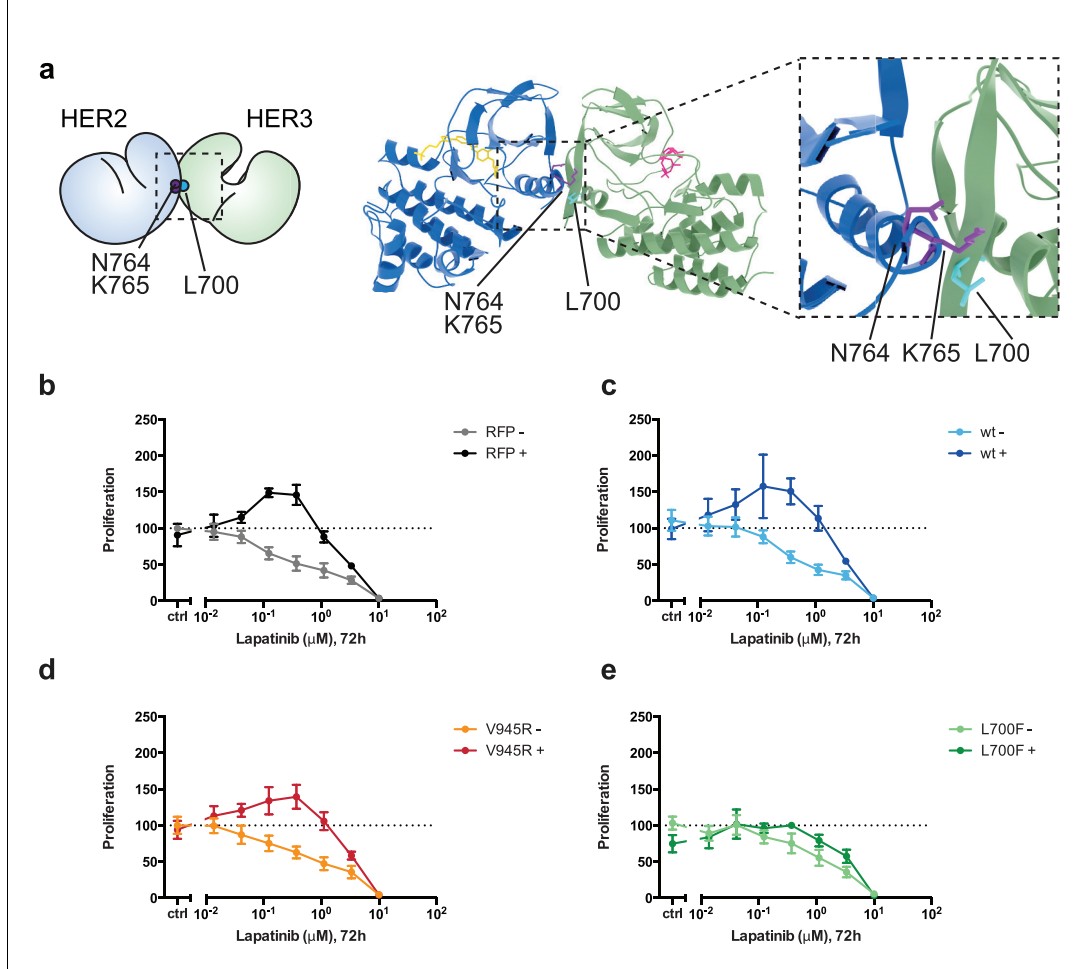

**Figure 6.** Disruption of the lapatinib-induced dimer inhibits lapatinib-NRG synergistic growth. (**a**) Molecular model of the lapatinib-induced HER2-HER3 dimer with the lapatinib-dimer interface residues HER2[N764] and HER2[K765] highlighted (purple), and a potential reciprocal residue HER3[L700F] (cyan). (**b–e**) 2D proliferation assays of SKBR3 cells transfected with (**b**) RFP empty vector, (**c**) HER3[wt], (**d**) HER3[V945R], or (**e**) HER3[L700F] and treated with lapatinib ±10 nM NRG as before. Data represents mean ±SEM for six independent experiments, each performed in triplicate. Corresponding data and statistics available as *Figure 6—source data 1*.

DOI: https://doi.org/10.7554/eLife.32271.031

The following video and source data are available for figure 6:

**Source data 1.** Numerical data and statistics relating to *Figure 6*.

DOI: https://doi.org/10.7554/eLife.32271.033

**Figure 6—video 1.** Interface highlight of the molecular model of a lapatinib-induced HER2-HER3 heterodimer in the HER3-like conformation, with HER2[I714] and HER3[V945] highlighted, as well as model-specific interface residues HER2[N764/K765] and HER3[L700].

DOI: https://doi.org/10.7554/eLife.32271.032

structural role that is critical to HER3 function, and apparently independent of any retained catalytic activity. Observing increased heterointeractions and cellular proliferation due to inhibition of an activity-deficient kinase is a strong indication of the importance of ATP-binding in certain pseudokinases, and the necessity of pocket-occupied structural conformers in sustaining protein–protein interactions and subsequent downstream signalling output.

Because of the importance of HER3 in HER2-targeted therapy resistance, its conserved ATP binding raised the possibility of targeting HER3 with ATP-competitive kinase inhibitors. Our data show, however, that stabilisation of the HER3 kinase domain with an ATP-competitive kinase inhibitor can have a stimulating effect on HER2[+] breast cancer cell proliferation. This indicates that the development of small molecule targeted therapy against HER3 for use in HER2[+] breast cancer needs to be directed away from stabilising the HER3 ATP binding pocket occupied conformer and rather towards

stabilising the apo, inactive conformer. An exception to this might be the development of irreversible, adamantane-linked inhibitors of HER3 that target the receptor for proteosomal degradation (*Xie et al., 2014*).

The substantial effect that lower doses of lapatinib have on proliferation in the presence of growth factor may have an impact on the establishment of lapatinib-resistance *in vivo*. This is in accordance with the observation from xenograft models that resistance occurs much more readily if lapatinib is administered continuously at low doses than if it's used intermittently at high dose (*Amin et al., 2010*). Increased production of growth factors (including NRG) is a well-described resistance mechanism against HER2-targeted therapy (reviewed in [*Claus et al., 2014*]). NRG production by the microenvironment has also been shown to play a role in metastatic spread of ovarian cancer cells that express high levels of HER3 (*Pradeep et al., 2014*). High expression levels of NRG in HER2$^+$ breast cancer patients showed a strong correlation with disease recurrence (*Xia et al., 2013*). Several somatic mutations in HER3 observed in cancer fall within the extracellular domain and have a potential effect on ligand-binding affinity (*Jaiswal et al., 2013*). These mutations may exacerbate the inhibitor-growth factor synergistic behaviour reported here.

Our results provide a potential molecular mechanism for the disappointing results observed in a recent Phase III study of lapatinib used in an adjuvant setting (ALTTO trial) (*Piccart-Gebhart et al., 2016*). The lapatinib-only arm of this study was terminated prematurely, and the effects observed in the adjuvant setting for both lapatinib-trastuzumab co-treatment and trastuzumab treatment followed by lapatinib were not significant. These clinical results indicate there are complicating factors in hindering lapatinib efficacy in patients, which may involve the expression levels of HER3 and NRG stimulation by a complex tumour microenvironment. The complex relationships between distinct protein conformation dynamics, formation of oligomeric assemblies, the availability of ligand, and the various effects on downstream signalling all need to be considered when applying targeted therapy to avoid potentially unexpected enhanced cancer cell proliferation after inhibitor treatment.

## Materials and methods

### Reagents and antibodies

NRG1 was purchased from PeproTech. Lapatinib was a kind gift from Professor György Kéri (Vichem Chemie Research Ltd Hungary). Bosutinib was purchased from LC Labs. Total HER2, HER3, PKB, HER2 pY877, HER3 pY1289, PKB pS473 and ERK1/2 pT202/pY204 antibodies were purchased from Cell Signaling Technology, anti-α-tubulin from Sigma, total ERK1/2 from Merck, and Alexa Fluor-488 conjugated anti-HER3 antibody from R&D systems.

### Cell culture and plasmid transfection

MCF7 and ZR75 cells were cultured in DMEM supplemented with 10% FCS, SKBR3 cells were grown in McCoys medium supplemented with 10% FCS. BT474, AU565, HCC1419, and HCC1569 cells were grown in RPMI with 10% FCS. For BT474 cells 10 μg/ml bovine insulin was included in the culture medium.

Cells were transfected with plasmid DNA using FuGENE 6, FuGENE HD (Roche), or Lipofectamine LTX (Thermo Fisher Scientific) according to the manufacturer's protocol.

All cell lines were sourced from the Francis Crick Institute's Cell Services facility, where they were tested negative for mycoplasma and authenticated via STR profiling.

### Proliferation assays

For 2D proliferation assays, cells were plated at $1 \times 10^4$ cells/well in a 96-well plate. The following day they were subjected to treatment for 72 hr, followed by addition of CellTiter-Glo reagent (Promega) and measured on an EnVision plate reader (Perkin Elmer). CellTiter-Glo data were normalised to the growth factor-null/inhibitor-null untreated control. This caused some growth factor-treated plots to start at above-baseline levels, which is an indication of the proliferative effect that growth factor treatment had in these cells.

For 3D spheroid assays, $3 \times 10^3$ cells were plated in a 96-well, round-bottom, ultra-low attachment plate (Corning) in the presence of 1% Matrigel (Corning). After three days of growth, an equal volume of 2x media containing treatment conditions was added and refreshed every three days for a

Cancer Biology | Cell Biology

total of eight days of treatment. Phase contrast images were taken using a Zeiss Axiovert 40 CFL microscope with a Zeiss 5x A-plan objective and analysed using ImageJ.

## Flow cytometry

SKBR3 cells were transfected with RFP-HER3 mutants for 48 hr. Cells were pre-treated with 0.5 mM EDTA to facilitate removal from the substrate and stained for HER3 extracellular expression using Alexa Fluor-488 conjugated anti-HER3 antibody (R&D systems, clone 66223) as per manufacturer's instructions. Briefly, cells were blocked using mouse IgG (Santa Cruz Antibodies) for 15 min at room temperature, followed by incubation with conjugated antibody for 30 min at room temperature in the dark. Cells were washed in PBS, 0.5% BSA, 0.1% sodium azide three times before flow cytometric analysis using a BD Fortessa instrument (BD). Results were analysed using the Flo-Jo software.

## FRET determination by FLIM measurements

Fluorescence resonance energy transfer (FRET) is used to quantitate direct protein–protein interactions and post-translational modifications. Processing of cells for FRET determination by FLIM has been previously described (*Barber et al., 2009*; *Parsons and Ng, 2002*). FLIM was performed using time-correlated single-photon counting (TCSPC) with a multiphoton microscope system as described previously (*Peter et al., 2005*). For experiments measuring endogenous protein, FRET pairs were Cy5-conjugated anti-HER2 IgG, and Alexa546-conjugated anti-HER3 IgG. For exogenous protein measurements, FRET pairs were HER2-GFP and HER3-HA with an anti-HA IgG, tagged with a Cy3 fluorophore. FRET efficiency between the donor and acceptor bound proteins was calculated with the following equation in each pixel and averaged per cell: FRET eff = 1-tau(DA)/tau(control) where tau(DA) is the lifetime displayed by cells co-expressing the donor and acceptor, whereas tau (control) is the mean donor (GFP) lifetime, measured in the absence of the acceptor.

## Modelling HER2-HER3 dimers

We modelled the HER2-HER3 dimer by comparative homology modelling using a multiple templates approach. The active, asymmetric HER2-HER3 dimer was modelled using the crystal structure of the active EGFR kinase domain (PDB ID 2GS2) (*Zhang et al., 2006*) and one chain of the crystal structure of the HER3 homodimer (PDB ID 3KEX) (*Jura et al., 2009b*) as templates. To build the EGFR-like, inactive, symmetric dimer we have used the crystal structure of the EGFR homodimer (PDB ID 3GT8) (*Jura et al., 2009a*), the crystal structure of EGFR complexed with lapatinib (PDB ID 1XKK) (*Wood et al., 2004*) and only one chain of the crystal structure of the HER3 homodimer (PDB ID 3KEX) (*Jura et al., 2009b*). To build the HER3-like dimer, we have used the HER3 homodimer structure (PDB ID 3KEX) (*Jura et al., 2009b*), the crystal structure of EGFR lapatinib-bound (PDB ID 1XKK) (*Wood et al., 2004*) and the crystal structure of the inactive EGFR AMP-PNP bound (PDB ID 2GS7) (*Zhang et al., 2006*). The sequence alignment used to build the model has been created by using PRALINE with the homology-extended alignment strategy (*Simossis et al., 2005*). We generated 200 three-dimensional models using the MODELLER package (*Sali and Blundell, 1993*). The selected models were chosen on the basis of the MODELLER objective function's DOPE score.

The volume of the HER2 ATP binding pocket was calculated with the SURFNET 1.5 package (*Laskowski, 1995*), where the cavity regions in a protein are built up by fitting a probe sphere of 1.4 Å$^3$ into the spaces between atoms.

The structural alignment was performed using the multi-seq tool of the VMD 1.9.1 package (*Humphrey et al., 1996*), and measurement of interaction surface buried residues was performed using POPScomp (*Kleinjung and Fraternali, 2005*).

## Receptor clustering assays

STORM imaging and cluster analyses are described in more detail at Bioprotocol (*Roberts et al., 2018*). SKBR3 cells were treated with either 14 nM Lapatinib or 41 nM Bosutinib. HER2 and HER3 Affibody ligands were used to label the non-activated states of the receptors (HER2 from Affibody Inc. and plasmid encoding the HER3 affibody was a gift from John Löfblom, protein made in house and shown to bind specifically to HER3 receptors) and NRG-β1 (Peprotech) was used to stimulate the cells. The conjugation of dyes (Invitrogen) to HER2 and HER3 ligands was done in house and the ratio of dye:ligand was confirmed to be ~1:1. The NRG-dye conjugate has been shown to be as

active as the unlabelled protein. We incubated cells in 100 nM HER2Affibody-Alexa488 + 50 nM HER3Affibody-Alexa647 or 100 nM HER2Affibody-Alexa488 + 10 nM NRG-Alexa647 ±drug for 1 hr. Cells were chemically fixed using 4% paraformaldehyde (EMS solutions)+0.5% glutaraldehyde (Sigma-Aldrich) diluted into ice-cold PBS.

Samples were imaged using a Zeiss Elyra super-resolution microscope to stochastically excite the Alexa488 and Alexa647 fluorophores bound to the receptors in the cells and to image single molecules. Imaging was done in TIRF mode using a 100x oil immersion objective lens. We used a 405 nm laser line to aid fluorophore blinking and 488 nm or 640 nm laser lines to excite the fluorophores, alternating the lasers to image the two receptors independently every 300 frames, over a total of ~10,000 frames. The exposure time was 20 ms. A minimum of two replicates of each sample were imaged generating at least 12 regions (25.6 µm x 25.6 µm) covering at least one cell per region. The Zen software localised the single molecule spots in the cells, a threshold was set to discard background spots and the co-ordinates of the positive localisations (typically 30,000 + for HER2 and 5,000 + for HER3 per region) were passed into the Bayesian cluster identification algorithm (*Rubin-Delanchy et al., 2015*)

The clustering algorithm expects the background and clusters to be uniformly distributed over a rectangular ROI (*Rubin-Delanchy et al., 2015*). The analysed images mainly showed single cells. Of interest are the HER2 and HER3 receptors in the cell membrane, which were visible as a circular shape. In order to conform with the prerequisites of the clustering algorithms, rectangular regions have been manually selected that tightly cover the cell membrane using the most suitable angles (assessed by visual inspection). The whole cell membrane has been covered in this way. The data selected by these regions have been rotated so that the sides of the rectangles became parallel to the coordinate axis. The result was used as input for the clustering algorithm and the algorithm was applied as described by the protocol. The complete lists of molecules per cluster that have been produced by the algorithm were used for the presentation.

## Recombinant HER3 KD purification

The baculoviral HER3 kinase domain construct was kindly provided by Prof. Mark Lemmon, University of Pennsylvania. Sf21 cells at $1 \times 10^6$ cells/ml were infected with P3 virus ($7 \times 10^7$ pfu/ml) at an MOI of 1.0 and allowed to grow for three days. The cells were lysed in lysis buffer containing protease inhibitors, 1 mM DTT and 2 mM BME. The lysate was clarified by centrifugation and incubated with NiNTA resin (Qiagen) for 30 mins at 4°C, after which the resin was washed extensively with buffer containing 50 mM Hepes (pH 7.6), 300 mM NaCl, 2 mM BME, 5% glycerol, 10 mM imidazole. HER3 was eluted in the same buffer with 200 mM imidazole added.

Each elution was centrifuged at 10,000 rpm to remove any precipitate or resin and applied to a S200 gel filtration column in 50 mM HEPES (pH 7.6), 300 mM NaCl, 2 mM BME, 2.5% glycerol.

## Thermal shift assay (TSA)

Thermal shift assays were carried out as described in (*Niesen et al., 2007*). Briefly, in a 96-well RT-PCR plate (Life Technologies) 1 µg HER3 kinase domain/well was incubated with 1 µM inhibitor or 200 µM ATP/10 mM $MgCl_2$ (as indicated) for 30 mins at 4°C in the presence of Sypro Orange dye (Sigma). HER2 TSA experiments were performed in a 384-well RT-PCR plate (Thermo Fisher Scientific). 0.5 µg of HER2 kinase domain/well was incubated with 1 µM lapatinib, 1 µM bosutinib, or 200 µM ATP/10 mM $MgCl_2$ for 20 mins at 4°C. HER3 measurements were taken on an Applied Biosystems 7500 Fast Real-Time PCR machine, and HER2 measurements on an Applied Biosystems Quant Studio 7 PCR machine. Data were trimmed and a Boltzmann sigmoidal curve fitted in GraphPad Prism 6. The inflection point of the Boltzmann sigmoidal was taken as the $T_m$. Thermal shift $\Delta T_m$ values were obtained by subtracting the $T_m$ value of the kinase domain alone control.

## Western blot analysis

Cells were plated at $0.5 \times 10^5$ cells/well in 24-well plates. Cells were lysed in 1x sample buffer (containing 1 mM DTT), sonicated and centrifuged. After centrifugation, the lysates were subjected to SDS-PAGE and analyzed by western blotting.

## Cellular thermal shift assay (CETSA)

CETSA was performed with COS7 cells transfected with HER3$^{wt}$-RFP, HER3$^{T787M}$-RFP or HER3$^{KGG}$-RFP plasmids as described in (*Jafari et al., 2014*; *Reinhard et al., 2015*). Briefly, COS7 were treated with DMSO or 50 nM bosutinib for 1 hr at 37°C. Cells were washed with PBS, detached and washed again twice with cold PBS. Cell pellets were resuspended in cold PBS with protease inhibitors (Roche) and 100 µl of each cell suspension was transferred into 0.2 ml PCR tubes. PCR tubes were heated for 3 min at 42°C or 50°C in a thermal cycler (DNA Engine DYAD, MJ research, Peltier thermal cycler) and incubated at room temperature for 3 min. Tubes were then immediately transferred onto ice, 35 µl of cold PBS 1.4% NP-40 with protease inhibitors were added and tubes were snap-frozen. Samples were then subjected to two freeze-thaw (at 25°C) cycles and cell lysates were centrifuged at 20,000 g for 1 hr at 4°C. Supernatants were carefully removed and analysed by western blot.

## Acknowledgements

We would like to thank Mark Lemmon for kindly providing HER3 baculoviral constructs. We wish to thank Melanie Keppler for making some of the HER2 mutants used in the experiments; as well as Luis Fernandes for some of the initial HER2-HER3 modelling work. We thank Gilbert Fruwirth for the fluorescent labelling of antibodies for the FRET-FLIM experiments. We thank the Flow Cytometry core facility at the Francis Crick Institute for carrying out the flow cytometry experiments and analysis. This work was supported by Cancer Research UK (C1519/A10331, C133/A1812, and C1519/A6906), the Biotechnology and Biological Sciences Research Council (BB/G007160/1 and BB/H018409/1), Dimbleby Cancer Care, KCL-UCL Comprehensive Cancer Imaging Centre (supported by Cancer Research UK/EPSRC) and in association with the MRC and DoH, The Medical Research Council (MR/L01257X/1 and MR/K015591/1), EU FP7 IMAGINT (EC GRANT: 259881), and the Swiss National Science Foundation.

## Additional information

### Funding

| Funder | Grant reference number | Author |
|---|---|---|
| Biotechnology and Biological Sciences Research Council | BB/G007160/1 | Marisa L Martin-Fernandez Peter J Parker |
| Medical Research Council | MR/K015591/1 | Marisa L Martin-Fernandez |
| Medical Research Council | MR/L01257X/1 | Franca Fraternali |
| Biotechnology and Biological Sciences Research Council | BB/H018409/1 | Franca Fraternali |
| Cancer Research UK | C1519/A6906 | Tony Ng |
| Dimbleby Cancer Care | Richard Dimbleby Professorship | Tony Ng |
| Cancer Research UK | C1519/A10331 | Tony Ng |
| Engineering and Physical Sciences Research Council | C1519/A10331 | Tony Ng |
| European Commission | 259881 | Tony Ng |
| Cancer Research UK | C133/A1812 | Tony Ng |
| Cancer Research UK | C1519/ A16463 | Tony Ng |
| Engineering and Physical Sciences Research Council | C1519/ A16463 | Tony Ng |
| Medical Research Council | C1519/ A16463 | Tony Ng |
| Department of Health | C1519/ A16463 | Tony Ng |
| Cancer Research UK | Institute core funding | Peter J Parker |

| Barts Cancer Institute | Institute core funding | Angus Cameron |

The funders had no role in study design, data collection and interpretation, or the decision to submit the work for publication.

## Author contributions

Jeroen Claus, Conceptualization, Formal analysis, Validation, Investigation, Visualization, Methodology, Writing—original draft, Writing—review and editing; Gargi Patel, Conceptualization, Formal analysis, Investigation, Methodology, Writing—review and editing; Flavia Autore, Gregory Weitsman, Francesca Collu, Formal analysis, Investigation; Audrey Colomba, Tanya N Soliman, Selene Roberts, Laura C Zanetti-Domingues, Formal analysis, Investigation, Writing—review and editing; Michael Hirsch, Formal analysis, Methodology; Roger George, Elena Ortiz-Zapater, Resources, Methodology; Paul R Barber, Boris Vojnovic, Resources, Software, Methodology; Yosef Yarden, Resources, Writing—review and editing; Marisa L Martin-Fernandez, Franca Fraternali, Tony Ng, Peter J Parker, Conceptualization, Resources, Supervision, Writing—review and editing; Angus Cameron, Conceptualization, Supervision, Writing—review and editing

## Author ORCIDs

Tanya N Soliman (ID) https://orcid.org/0000-0002-4687-629X
Selene Roberts (ID) http://orcid.org/0000-0002-3732-0556
Paul R Barber (ID) http://orcid.org/0000-0002-8595-1141
Peter J Parker (ID) http://orcid.org/0000-0002-6218-2933

## Decision letter and Author response

Decision letter https://doi.org/10.7554/eLife.32271.036
Author response https://doi.org/10.7554/eLife.32271.037

# Additional files

## Supplementary files

• Transparent reporting form
DOI: https://doi.org/10.7554/eLife.32271.034

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
