## [Decision Letter]

Thank you for submitting your article "Inhibitor-induced HER2-HER3 heterodimerisation promotes proliferation through a novel dimer interface" for consideration by *eLife*. Your article has been reviewed by two peer reviewers, and the evaluation has been overseen by a Reviewing Editor and Jonathan Cooper as the Senior Editor. The following individual involved in review of your submission has agreed to reveal his identity: Nathanael S Gray (Reviewer #2).

The reviewers have discussed the reviews with one another and the Reviewing Editor has drafted this decision to help you prepare a revised submission.

The reviewers were generally quite impressed by the manuscript, noting that it represented a very nice molecular pharmacology detective story to try to explain the paradoxical activation of Her2 signaling upon treatment with an ATP-competitive Her2 inhibitor Lapatinib. Key observations supported by the data presented include 1) the Her2/Her3 dimer induced by lapatinib + neuregulin does not resemble the head to tail model for "normal" activation by neuregulin alone, but a head to head orientation, previously only observed in the HER3 crystal structure 2) the Her3 ATP site needs to be occupied by a ligand (ATP or bosutinib) for the hyperactivation to occur, 3) an irreversible version of lapatinib (neratinib) does not activate cell growth – presumably because the compound is irreversibly bound to HER2. The reviewers felt that these observations are an important contribution to our understanding of HER2/HER3 signaling in breast cancer. However, several points were raised for revision, including some that will require additional experiments.

Essential revisions:

1) The authors use various mutations in Her3 to show ATP binding by Her3 is important to maintain activating ability of lapatinib. They should show CETSA or other binding data for what these mutations do to bosutinib affinity of Her3 in cells. Also I wondered what bosutinib combined with lapatinib does in the presence of these mutations (synergy or antagonism and do the mutations abrogate this effect)? The authors could also have used the covalent Her3 binder TX1-85-1 (Nat Chem Biol. 2014 Dec; 10(12): 1006-1012.) which is more selective than bosutinib as a way to interrogate what Her3 ATP-site ligation does to the pharmacology of lapatinib.

2) Much of the mechanistic analysis is done in SKBR3 cells, which from the data presented appear to be somewhat of an outlier in terms of the Neuregulin + lapatinib induction of growth. In Figure 1—figure supplement 1 the authors investigate three other cells, BT474, MCF7, and ZR75. The first is a high Her2 expresser and also shows the same Neuregulin + lapatinib enhanced growth. Several more HER2 overexpressing cell lines should be tested for this effect (2-5 should suffice).

---

## [Author Response]

Essential revisions:1) The authors use various mutations in Her3 to show ATP binding by Her3 is important to maintain activating ability of lapatinib. They should show CETSA or other binding data for what these mutations do to bosutinib affinity of Her3 in cells. Also I wondered what bosutinib combined with lapatinib does in the presence of these mutations (synergy or antagonism and do the mutations abrogate this effect)? The authors could also have used the covalent Her3 binder TX1-85-1 (Nat Chem Biol. 2014 Dec; 10(12): 1006-1012.) which is more selective than bosutinib as a way to interrogate what Her3 ATP-site ligation does to the pharmacology of lapatinib.

Based on the reviewer’s suggestions we have included CETSA binding data of COS7 cells transfected with HER3^wt^, HER3^KGG^ or the bosutinib-desensitised mutation HER3^T787M^. We compared the recovery of HER3 protein between 44°C and 50°C in the presence or absence of bosutinib and show that bosutinib stabilises HER3^wt^, but it does not stabilise either the HER3^KGG^ or HER3^T787M^ mutations. This corresponds to our proliferation data to suggest that HER3^KGG^ does not have strong affinity for bosutinib in cells. This data is included in the revised manuscript as Figure 2—figure supplement 1G.

The reviewers suggest the use of the covalent HER3 inhibitor TX1-85-1 as a more selective inhibitor than bosutinib. In the original publication cited (Xie et al., 2014) the authors demonstrate TX1-85-1 potently inhibits HER2 (Supplementary Table 1 in Xie et al.). Although the covalent binding to HER3 would make TX1-85-1 an interesting tool to probe ATP-binding pocket behaviour in HER3, the fact that it is also a potent HER2 binder would confound the analysis in the proposed assays.

HER3 has been reported to be the only EGFR family member that tightly binds bosutinib due to an alanine equivalent to Src^A403^, which is lacking in other EGFR family members (Levinson et al. Nat Chem Biol 2013, DOI 10.1038/NCHEMBIO.1404). This is supported by screening data from Davis et al., 2011. To verify, we have included *in vitro* TSA analysis of HER2 kinase domain treated with Mg^2+^/ATP, lapatinib or bosutinib. This data agrees with the published results on HER2-bosutinib binding, showing that HER2 is not a strong binder of bosutinib in our hands. This data is included in the revised manuscript as Figure 2B.

2) Much of the mechanistic analysis is done in SKBR3 cells, which from the data presented appear to be somewhat of an outlier in terms of the Neuregulin + lapatinib induction of growth. In Figure 1—figure supplement 1 the authors investigate three other cells, BT474, MCF7, and ZR75. The first is a high Her2 expresser and also shows the same Neuregulin + lapatinib enhanced growth. Several more HER2 overexpressing cell lines should be tested for this effect (2-5 should suffice).

As the reviewers suggested, we have included additional cell lines in the revised manuscript and reclassified the presentation of the various cell lines in Figure 1—figure supplement 1. Instead of referring to the HER2 expressions status, we present the data as ‘lapatinib-responding’, ‘partial lapatinib response’ and ‘lapatinib non-responding’, based on the cellular toxicity presented by lapatinib. We match the phenotype seen in SKBR3 with BT474 and AU565, show a partial match with ZR75 and HCC1419, and a negative response with MCF7 and HCC1569.